# The Use of Medical Ozone in Chronic Intervertebral Disc Degeneration Can Be an Etiological and Conservative Treatment

**DOI:** 10.3390/ijms24076538

**Published:** 2023-03-31

**Authors:** Anibal Martin Grangeat, Maria de los Angeles Erario

**Affiliations:** Instituto Argentino de Ozonoterapia (IAOT), Ciudad Autonoma de Buenos Aires C1425ASG, Argentina

**Keywords:** ozone, oxidative stress, intervertebral disc, inflammation, Nrf2, ROS, macrophages, mesenchymal stem cells, low back pain, hormesis

## Abstract

Degeneration of the intervertebral disc is one of the most frequent causes of lumbar pain, and it puts an extreme strain on worldwide healthcare systems. Finding a solution for this disease is an important challenge as current surgical and conservative treatments fail to bring a short-term or long-term solution to the problem. Medical ozone has yielded excellent results in intervertebral disc pathology. When it comes to extruded disc herniation, ozone is the only etiological treatment because it stimulates the immune system to absorb the herniated portion of the nucleus pulposus, thus resolving discal extrusion. This work aims to examine the biomolecular mechanisms that lead to intervertebral disc degeneration while highlighting the significance of oxidative stress and chronic inflammation. Considering that ozone is a regulator of oxidative stress and, therefore, of inflammation, we assert that medical ozone could modulate this process and obtain inflammatory stage macrophages (M1) to switch to the repair phase (M2). Consequently, the ozone would be a therapeutic resource that would work on the etiology of the disease as an epigenetic regulator that would help repair the intervertebral space.

## 1. Low Back Pain

Low back pain (LBP) is a common condition affecting approximately 637 million individuals worldwide, resulting in high socioeconomic costs, and it is the fifth most common reason for seeking healthcare help. Intervertebral disc degeneration (IVDD) is a major contributor to LBP. It is a serious public health problem, and it has been identified as the single most common cause of disability worldwide by the 2013 Global Burden of Disease Study (Collaborators of Global Burden of Disease Study, 2015). LBP has been reported as the biggest contributor to the global Years Lived with Disability (Collaborators of Global Burden of Disease Study, 2015). It has a lifetime prevalence of over 80% in North America, and it is estimated that in the United States alone, LBP costs about 100 billion dollars per year for health care services and paid medical leave, resulting in a tremendous socio-economic burden [1,2].

## 2. Intervertebral Disc

The intervertebral disc (IVD) is composed of the central nucleus pulposus (NP), surrounded by the annulus fibrosus (AF) and hyaline cartilaginous endplates (CEP), which encompass these two structures at the junction of the vertebrae. The IVD is an avascular structure and is, therefore, reliant on concentration gradients for the diffusion of nutrients and oxygen through the adjacent endplates [3].

### 2.1. Anatomy of the Intervertebral Disc

#### 2.1.1. Nucleus Pulposus

The central NP contains collagen fibers (Type II), which are organized randomly, and elastin fibers arranged radially; these fibers are embedded in a highly hydrated aggrecan-containing gel. Aggrecan is the major proteoglycan (PG) in the NP. Proteoglycans (PGs) are macromolecules that are composed of a protein body covalently linked to sulfated polysaccharide chains called glycosaminoglycans (GAGs). These sulfated GAGs carry an overall negative charge that is responsible for the retention of water molecules efficiently, allowing hydration of the extracellular matrix (ECM). Aggrecan has a major role in maintaining disc height and turgor against compressive loads by attracting water molecules inside the NP extracellular matrix. Notably, this water content, 77%, along with type II collagen fibers, enables the NP to be elastic and deformed under stress. PGs to type II collagen ratio is 27:1 [4]. It also contains other collagen types in smaller amounts and has important functions in maintaining the characteristics of the hydrogel matrix [5]. Cells of the NP are highly specialized and survive in a very hypoxic environment (1–2% of O_2_). Therefore, NP cells possess a unique metabolism that allows them to be functionally and constitutively adapted to their environment, which is low in oxygen and nutrients [6]. Cells in the NP at birth are predominantly of notochordal origin, but they become almost undetectable in the first decade of life. At this point, the NP region becomes gradually populated, especially in regions close to the CEPs, by a significant majority of chondrocyte-like rounded cells. The NP contains approximately 3000 cells/mm^3^ [7].

#### 2.1.2. Annulus Fibrosus

The NP is surrounded by the annulus fibrosus (AF), which prevents the extrusion of the NP when excessive loads are applied to the IVD [8]. The complex AF structure is often separated into two distinct regions: the inner AF containing primarily type II collagen and the outer AF containing primarily type I collagen. Four distinct types of AF cells reside in the normal, healthy AF: peripheral cells, outer annulocytes, inner annulocytes, and interlamellar cells. AF stem cells and neuronal stem cells in healthy AF tissue have also been identified. From the outer to inner AF regions, the ratio of type I collagen to type II collagen decreases. In the human adult, reported AF cell density ranges from approximately 3000 cells/mm^3^ to 9000 cells/mm^3^. The water content of AF is 70%. From the outer AF region towards the inner AF, there is a marked change in cell morphology and phenotype to a more rounded cell shape, chondrocyte-like cell [9]. 

#### 2.1.3. Cartilage Endplates

The third morphologically distinct region is the cartilage endplate (CEP) of hyaline cartilage. Cells within the CEP are similar to articular chondrocytes, which synthesize an extracellular matrix ECM that is rich in type II collagen and PGs. In these plates, the ratio of PG to type II collagen is approximately 2:1, and the water content is 50–60%. CEPs are also the primary source of nutrition for the IVD. The bony component of the end plate has a structure, not unlike that of the vertebral cortex and resembles a thickened, porous layer of fused trabecular bone with osteocytes entombed within saucer-shaped lamellar packets. The vertebral endplates also have the unique role of acting as the main transport for nutrients in and out of the disc. There are specialized capillary beds between the vertebral body and endplate that allow the passive diffusion of nutrients and oxygen into the intervertebral disc [10].

### 2.2. Physiology of the Intervertebral Disc

The IVD is avascular, and there are, thus, steep gradients in the concentration of nutrients and metabolites from the blood supply at the disc’s margins and to the cells in the center of the disc. These gradients form when the cells use nutrients and produce metabolites at rates that are high compared to rates of transport into the disc and diffusion through the tissue. The actual concentrations of glucose, oxygen, and lactic acid in the disc will, thus, depend both on parameters affecting nutrient transport and on cellular metabolic rates [11]. Therefore, a flow of nutrients occurs via diffusion from the pre-disc vessels that reach into the outermost layers of the disc. A healthy IVD is also considered a poorly innervated organ. Innervation is restricted to the outer layers of the AF and consists of sensory and sympathetic perivascular nerve fibers [12,13]. This unique physiological structure without significant innervation and vascularization has led to the consideration that IVD is an immune-privileged organ [14].

## 3. Intervertebral Disc Degeneration

Adams and Roughley have proposed a useful definition of IVD degeneration (IVDD) as “the process of disc degeneration is an aberrant, cell-mediated response to progressive structural failure” [15]. The IVD differs from other connective tissues in the body in that it shows degenerative and aging conditions early in life. Age-associated maturation of IVDs, suggestive of degeneration, has been observed as early as age 11, and histological evidence of diminished blood supply to the vertebral endplates has been reported to start in the second decade of life with increasing prevalence with advancing age [16].

### 3.1. Etiology of Intervertebral Disc Degeneration

Intervertebral disc degeneration (IVDD) often occurs when the catabolism of the extracellular matrix overcomes the anabolism [17]. Different predisposing factors influence IVDD: genetic factors, age, persistent mechanical stress, smoking, unhealthy lifestyle, occupational exposures, trauma, and possible infective microorganism infiltration after previous minor trauma. Genetic inheritance is the most prevalent risk factor associated with the development of IVDD, which was illustrated and confirmed by the analysis of several genes [1,18]. After genetic predisposition, the principal cause of disc degeneration is the altered nutrient supply to the disc cells. A fall in nutritional supply causes a fall in oxygen quantity and a rise in lactate concentration with consequent pH alteration, affecting the cell function and synthesis of the extracellular matrix. In the long term, this leads to a degenerative process [17].

### 3.2. Morphological Changes in Intervertebral Disc Degeneration 

Morphologic degeneration may include narrowing of the disc space, fibrosis, diffuse bulging of the annulus beyond the disc space, annular tears and mucinous degeneration of the annulus, desiccation, defects, and sclerosis of the end plates. It is clinically manifested as painful syndromes (especially back pain) occurring in the elderly [19]. The NP shows the most extensive changes in IVDD, and it is, therefore, the most thoroughly investigated. Both the AF and CEPs have received attention in their relationship with IVDD; however, changes in these structures are less well documented. The degenerated NP is an unorganized fibrous tissue that has largely lost its capacity to bind water (by loss of PGs) under compression. Therefore, the intradiscal pressure in the NP is increased, and disc height is lost. AF deforms and shows a progressive increase in structural defects. There is an increase in microscopic and macroscopic damage to the endplate and a marked increase in sclerosis of the subchondral bone [20]. Although disc herniation is most commonly due to mechanical injury and consequent rupture of the AF, some extent of initial degeneration is necessary to allow the NP to herniate through fibrous bands of AF into the vertebral canal [21]. For a healthy disc to rupture, an enormous force is necessary. In many cases, the CEPs of the vertebrae fail sooner than the AF [22,23].

### 3.3. The main Degenerative Events

#### 3.3.1. Cells

The first signs of IVDD begin to appear upon skeletal maturity in the NP. Until this time, two cell types populate the NP: chondrocyte-like cells and notochordal cells. Notochordal cells are responsible for maintaining homeostasis, and the loss of these cells during skeletal maturation might constitute one of the first changes that occur in the cascade of degenerative events. Notochordal cells die by apoptosis when the sclerotome condenses and proliferates to form vertebral bodies [24]. As notochordal cells diminish after birth and are gradually replaced by smaller chondrocytes, the NP becomes dehydrated and cartilage-like by adulthood. Disc degeneration is an age-related process. It is difficult to distinguish the physiological process of disc aging from pathological disc degeneration. Generally, when a disc with structural failure is combined with accelerated or advanced signs of aging, it is considered to be a degenerated disc [25]. The significant pathological features of IVDD are characterized by progressive loss of active NP or AF cell numbers and altered phenotype of normal disc cells [26].

#### 3.3.2. Inflammatory Mediators

The secretion of inflammatory mediators tends to derive from the degenerative NP cells, as well as from circulating immune cells that infiltrate inside the IVD due to the favorable conditions generated during IVDD. The infiltration of activated immunocytes, including macrophages, T and B cells, and natural killer cells, occurs in response to the expression of a number of cytokines released by IVD cells, and it is allowed by the loss of structural integrity of the outer AF. The presence of inflammatory mediators producing immune cells within the IVD, which does not contain a resident immune cell population in physiological conditions, enhance inflammation inside the disc tissues [27,28]. Many studies have shown that inflammation factors play a key role in the process of IVDD. The interaction and abnormal expression of inflammatory factors can disrupt the balance of ECM metabolism, cause inflammation, and accelerate IVDD. Tumor necrosis factor (TNFα), interleukins (IL), nitric oxide, and prostaglandin E2 (PGE_2_) are the main factors for the inflammatory reaction in IVD [29]. Cytokines initiate the inflammatory cascade by stimulating a range of pathogenic responses in the disc cells that could promote autophagy, senescence, and apoptosis. It has been studied that expression of IL-1β, IL-1R, and TNFα are increased in degenerated disc tissue and mediates catabolic effects, decreasing proteoglycan production and enhancing matrix metalloproteinases (MMPs) expression. The degradation of ECM is mainly due to overexpression of MMPs leading to loss of aggrecan, collagen and dehydration of NP cells with impairment of mechanical function in the disc [20]. The most important pro-inflammatory and anti-inflammatory cytokines are described in Table 1.

#### 3.3.3. Oxidative Stress: A Trigger for Inflammation

The microenvironment of healthy discs is characterized by hypoxia (1–2% O_2_), low nutrition, and acidic pH due to lactic acid accumulation. Insufficient supplement of nutrients and the accumulation of metabolites together bring about a tough microenvironment in degenerative IVD. This special local microenvironment may serve as a trigger for the overproduction of reactive oxygen species (ROS) [31]. While the microenvironment of IVDs is characterized by hypoxia due to poor vascularization, all resident disc cells (NP cells, AF cells, and CEP cells) have been demonstrated to be not anaerobic and to have oxygen-utilizing metabolic processes in vivo. Therefore, disc cells are expected to produce ROS in the microenvironment of discs. Actually, H_2_O_2_ has been identified in human NP tissues [32]. The major site of ROS production in non-immune cells is in the mitochondria. The dysfunction of chondrocyte mitochondria is the primary site of ROS production. When fissures occur in the IVD, resulting in neovascularization, the disc cells become exposed to unusually high oxygen levels, which leads to a rise in ROS, leading to inflammation by oxidative stress. Production of ROS at high oxygen tension also correlates with reduced PG synthesis and enhanced expression of catabolic factors [33]. Meanwhile, both the contents of enzymatic antioxidants and non-enzymatic antioxidants are reduced, leading to impaired scavenging of ROS and can cause oxidative damage to DNA, lipids, and proteins and also enhance ROS production in disc cells, forming a positive feedback loop [31]. The intracellular redox homeostasis depends on a balance between ROS generation and ROS scavenging performed by nonenzymatic and enzymatic antioxidants, including glutathione (GSH), superoxide dismutase (SOD), catalase (CAT), glutathione peroxidase, ascorbic acid (vitamin C), α-tocopherol (vitamin E), and carotenoids. Finding a therapeutic resource that allows the capture of free radicals from the mitochondria could prove promising to delay the degeneration of the intervertebral disc.

ROS regulates apoptosis because they promote the diminution of functional IVD cells, accelerating degeneration. They also regulate autophagy and trigger senescence. Excess ROS causes oxidative stress to activate several signaling pathways in disc cells, including the nuclear factor kappa B (NF-κB) pathway and the Mitogen-activated protein kinase (MAPK) pathway. The signaling response to ROS is cell-type dependent. Consequently, the phenotype of the disc cells changes from an anabolic matrix phenotype to a proinflammatory and catabolic matrix phenotype. This change causes a drastic loss of matrix and increases inflammation in the microenvironment of the discs. Additionally, the cytokines secreted by disc cells recruit more immune cells to the discs to further increase inflammation. These immune cells secrete more cytokines to impair disc cell viability and function, forming a vicious cycle [32]. NF-κB is activated in response to numerous types of stress, including oxidative stress. Chronic activation of NF-κB is associated with numerous diseases, including. musculo-skeletal diseases. Growing evidence indicates that NF-κB becomes activated in aged tissues in response to accumulated damage and mediates the degenerative changes. Activation of the NF-κB signaling system occurs in the human IVD especially in the nucleus pulposus tissue [34] and is an indispensable molecule to modulate transcription of genes encoding proinflammatory cytokines, adhesion molecules, and chemokines, as well as growth factors and inducible enzymes, which is a possible treatment target for inflammatory disease therapy [35].

#### 3.3.4. Mitochondrial Dysfunction and Oxidative Stress: Vicious Cycle

Mitochondrial dysfunction in IVDD is also known to promote degeneration just by diminishing the adenosine triphosphate (ATP) production, which has a key role in maintaining the extracellular matrix integrity. Mitochondrial dysfunction can lead to abnormal mitochondrial morphology, decreased membrane potential, defection of respiratory chain, opening of mitochondrial permeability transition pore, and decline in mitochondrial membrane potential leading to an overproduction of ROS, inducing apoptosis of the NP, aging, and IVDD [36].

#### 3.3.5. Macrophages and IVDD

Macrophages have a large plastic phenotype and display distinct subtype changes and functional differences in different microenvironments. They are the only inflammatory cells that can penetrate the closed nucleus pulposus and their polarization plays an important role in IVDD and they were found in large numbers in the discs of patients with IVDD. M0 macrophages are non-activated, which can be activated in both classical activation (M1) and alternative activation (M2). M1 macrophages can secrete a variety of proinflammatory cytokines: TNF-α, IL-1β, IL-6, IL-12, and MCP-1, displaying notable pro-inflammatory effect and they have high bactericidal activity. The levels of these proinflammatory factors secreted by cells were found to be markedly elevated in IVDD, which promoted the degradation of extracellular matrix, facilitated the phenotype change of cells and resulted in an imbalance of catabolism and anabolism, thus leading to the occurrence of IVDD. M2 macrophages can secrete anti-inflammatory factors, such as IL-4, IL-1R, IL-10, TGF-β and PDGF, primarily playing anti-inflammatory and promoting tissue repair role [37]. M1 polarization of macrophages could promote inflammation and damage of NP cells and M2 polarization could be a therapeutic target [38]. Host IVD cells can be sensitized by chronic inflammatory signals and by the increased presence of macrophages that remain in pro-inflammatory and remodeling stages and do not revert to anti-inflammatory healing stages. It is broadly known that aged and degenerated human IVDs have particularly poor healing capacity due to low cellularity, accumulation of structural defects, and chronic inflammation. The remodeling phase of the wound can occur over a period of years in humans [9]. In summary, IVDD is a pathology with different predisposing etiological factors and where multiple biomolecular mechanisms are involved. These mechanisms are schematically detailed in Figure 1.

## 4. Treatments for IVDD

There exists no effective, surgical or non-surgical treatment for IVDD nowadays, which is largely related to the lack of knowledge of the specific mechanisms of IVDD, and the lack of effective specific therapeutic targets [29].

### 4.1. Conservative and Surgical Treatments for IVDD

In 2010 Janna Friedly et al. reviewed and analyzed the techniques of epidural steroid injections, Z-joint injections and RFN, emerging techniques, surgical interventions, physical therapy and exercise programs, and opioid use. They concluded that in most circumstances, the chronic state, cannot be successfully treated with individual interventions of any kind [39]. More recently Wu et al. conducted a comprehensive review of the therapeutic options available for the treatment of intervertebral disc degeneration. They classified the different treatments: treatment options for pain relief in conservative therapy, such as physical strengthening and physiotherapy, oral medications, pain-relieving injections. Treatment with the aim of restoration, repair, and regeneration of intervertebral disc: cell therapy, growth factor therapy, and gene therapy. Percutaneous intervertebral disc techniques: mechanical decompression, thermal decompression, chemical decompression, biomaterial implantation and finally surgical management [40]. IVDD is considered as a chronic inflammatory state. The non-surgical treatment only relieves the symptoms, and surgical treatment has the defects of recurrence, aggravation of adjacent segment degeneration and difficulty in restoring normal biological function of the spine [37]. Research groups have aimed to regulate the redox balance of disc cells as a therapeutic objective to alleviate oxidative stress. So, as the role of oxidative stress is a key point in the development and progression of intervertebral disc degeneration, having a therapy that regulates oxidative stress within the intervertebral space seems to be a potential therapeutic resource.

### 4.2. Medical Ozone 

In the last decades the medical use of ozone has been progressively increasing all over the world as a treatment for several diseases [41]. Ozone (O_3_) is a molecule composed of three oxygen atoms. Medical ozone applications date back to the beginning of the last century [42]. It is a highly unstable gas and it is clinically used as an oxygen-ozone gas mixture at low concentrations for multiple therapeutic purposes [43]. Ozone reacts with organic compounds containing double bonds and adds the three oxygen atoms to the unsaturated bond with the formation of ozonides. This reaction is of great importance, since ozone causes the split of the double bonds with a reaction called ozonolysis [44]. The mechanisms of action of ozone in mammalians, when used in adequate doses, are based on the fact that the brief and controlled oxidative stimulus leads to the formation of reactive oxygen species and lipid peroxides, which in turn act as second messengers. This is an apparently paradoxical concept: ozone could induce an antioxidant response capable of reversing transient oxidative stress, because after this oxidative stimulus there is an increase of antioxidant enzymes [45]. When ozone comes into contact with human fluids and tissues, it reacts with polyunsaturated fatty acids (PUFA), creating hydrogen peroxide (H_2_O_2_) and a mixture of lipid ozonation products (LOP), mainly composed by 4-HNE (from omega-6 PUFA) and 4-HHE (trans-4 hydroxy-2-hexenal from omega-3 PUFA). H_2_O_2_ is an important reactive oxygen species (ROS), and it acts as a messenger with extremely short lifetime (a few seconds) [46,47,48,49,50]. The moderate oxidative stress caused by ROS is counteracted by endogenous radical scavengers, such as superoxide dismutase, glutathione peroxidase, catalase, and NADPH quinone-oxidoreductase. It has been shown that small and repeated oxidative stresses could induce the activation of a transcriptional factor mediating nuclear factor erythroid 2 related factor 2 (Nrf2), a domain involved in the transcription of antioxidant response elements (ARE) [51,52,53]. Under basal conditions, Nrf2 is sequestered in the cytoplasm by its specific inhibitor Keap1 (Kelch-like ECH associated protein). Under specific stimuli, Nrf2 dissociates from Keap1, translocates into the nucleus and transactivates ARE-driven genes [54]. Low doses of ozone can increase the level of nuclear translocated Nrf2, which is associated with an increase in Nrf2 protein translocation from the cytoplasm to the nucleus. Consequently, Nrf2 increases the activity of the antioxidant and phase II detoxifying enzymes, which collectively favors cell survival [55].

## 5. Spinal Disease and Medical Ozone

### 5.1. Disc Herniation and Ozone

The efficacy of ozone therapy in the painful pathology of the spine has been widely demonstrated [56,57,58,59,60,61,62,63,64]. In the extruded disc herniation when the NP is exposed to the immune system, an antibody–antigen reaction takes place, and then an inflammatory response and inflammatory mediators cascade occur. This autoimmune mechanism tries to produce the reabsorption of the herniated portion. The process is slow and painful, and it never consolidates. Therapeutic doses of ozone produce activation of Nrf2-antioxidant signaling consequently attenuating activation of NF-kB, a key regulator of inflammation, therefore reducing pain. In these concentrations, ozone stimulates macrophages phagocytosis of the herniated portion and its complete resorption, preserving the nucleus pulposus vitalized within the intervertebral space, improving the quality of the final scar formation [65].

### 5.2. Intervertebral Disc Degeneration (IVDD) and Ozone

Through years of observation, treatment and experience we have observed the positive therapeutic effects of ozone in IVDD. With the progress of molecular biology studies, today we can justify these effects. Ozone improves IVDD for different reasons. 

#### 5.2.1. Ozone and ROS in IVDD

It has been demonstrated there is a significantly different status of oxidative stress between normal intervertebral disc tissues and IVDD tissues which indicates that immunity abnormality plays an important role in the pathogenesis of disc disease [66]. As a consequence of mitochondrial dysfunction there is an overproduction of ROS. At the same time, the function of antioxidant enzymes in the intervertebral disc is inhibited, and the ROS clearance rate decreases. This oxidative stress induces inflammation, dysfunction and phenotype changes of cartilage endplates, leading to NP cell nutritional metabolism disorders which impair the metabolic balance of ECM. This excess of ROS induces apoptosis leading to the decline of cell number, senescence and autophagy [67]. Moreover, tears in the AF produce an increase in permeability of the intervertebral disc. In this way, the IVD is invaded by granulation tissue with blood vessels and pathological innervation. The blood vessels bring cells from the immune system that release inflammatory mediators, generating even more ROS and promoting a vicious cycle: ROS-proinflammatory cytokines [68]. 

Different experimental antioxidative therapies have been proven in IVDD. For example, treatment with N-Acetylcysteine NAC could reverse the NP cell apoptosis and ECM degradation [69]. Resveratrol (RSV) could inhibit MMP-13 expression and promote proteoglycan synthesis in NP cells and could protect NP cells from degradation [70]. Pyrroloquinoline (PQQ) has the potential to eliminate ROS production and reduce the apoptotic process [71]. In the same way that antioxidant therapies are being studied and tested as a treatment for IVDD, ozone appears as an optimal therapeutic resource. According to widely studied properties ozone modulates oxidative stress through the activation of Nrf2 pathway in cells, and then induces the synthesis of cytoprotective proteins, which favors cell survival and improves the oxidative stress in IVDD. By this mechanism, could decrease the induction of apoptosis, autophagy and senescence. Autophagy may have a dual role; adequate autophagy may increase cell survival but excessive autophagy may increase cell death. That is why it is so important to determine the right doses of ozone used for the treatment for IVDD. Through this mechanism, the activation of NF Ϗβ decreases, controlling the inflammatory state in the intervertebral space avoiding the continuity of the vicious cycle that is established between oxidative stress and inflammation. This is why ozone could promote an improvement in the microenvironment of the intervertebral space, making it a favorable microenvironment for regenerative therapies (e.g., PRP, stem cells, growth factors, exosomes, progenitor cells, etc.). Nrf2 is cytoprotective of the NP cell.

#### 5.2.2. Ozone and Polarization of Macrophages

It has been demonstrated that macrophages are the only type of inflammatory cells infiltrated in the degenerated disc and correlated with disease progression [72]. It was traditionally thought that macrophages are mainly phagocytotic in function, although later work has demonstrated numerous additional functions of macrophages. Compared to the classical phagocytotic “M1” macrophages, the alternatively polarized macrophages, called “M2” macrophages, function as modulators of cellular and humoral immunity and as mediators of tissue repair and remodeling. Importantly, M2-like polarization induces fibrosis, mainly mediated by TGFβ signaling. M2 polarization of macrophages induced by epigenetic modulations may have a demonstrable therapeutic effect on IVDD [73]. IVDD is an inflammatory state, and ozone could act as an epigenetic modulator to induce polarization of macrophages M1 to M2 as in extruded disc herniation. It has been described that medical ozone decreased phagocytosis of polymorphonuclear (PMN) in healthy cows but increased in cows with mastitis and milk fever [74]. Moreover, Fernandez-Cuadros et al. [75] describes that ozone also reduces MMPs which have a catabolic effect on articular cartilage demonstrating that medical ozone modulates inflammation. 

### 5.3. The Concept of Ozone in Low Doses

Low doses of O_3_ might also have a role in the regulation of prostaglandin synthesis, the release of bradykinin, and in increasing secretions of macrophages and leukocytes [51]. Low ozone concentration stimulates cell protective pathways and nuclear transcription via the activation of an antioxidant response through induction of an oxidative “eustress” able to stimulate cell defense pathways without causing deleterious effects [54]. Costanzo et al. have demonstrated that the effects of ozone administration are dependent on gas concentration, and that the nucleus and the cytoplasmic organelles may be differently affected. They studied the direct effects of mild ozonization on some cellular mechanisms. They proved that ozone at 10 µg/mL induces positive and long-lasting cellular responses in cytoskeletal organization and mitochondrial activation, as well as in nuclear transcription in cell culture. On the other hand, 1 µg O_3_/mL-treatment seems to represent a stimulus able to activate some transient responses.The low-dose ozone concept with its moderate oxidative stress represents an ideal hormesis strategy. Dose-response and concentration-effect relationships in the context of specific applications allow one to fix concentration ranges with therapeutic benefit. In its pharmacological effect, medical ozone follows the principle of hormesis: low concentrations (or doses) show high efficacy, which decreases with increasing concentration [76,77].

## 6. Discussion

IVD degeneration is a progressive and invalidating disease involving high socioeconomic cost, with a multifactorial etiology and is the consequence of failure in tissue repair. It can begin to manifest clinically in three different ways. In its acute presentation, the extruded disc herniation (EDH) starts out as sharp and sudden LBP which in a matter of hours radiates to one or both lower limbs, with major pain in one of them. In its chronic form, it initiates with mild pain, which gradually intensifies and often inability to walk. In its mixed form, a mild progressive chronic pain, sharp leg pain may occur. This is due to the fact that during a degenerative disc disease (DDD) an EDH may occur. Even though various treatments have been proposed over time, the chief clinical practice guides and research groups agree that the hitherto available treatments do not offer a lasting solution for degeneration of the intervertebral disc. The reason these treatments fail is that they aim to solve the symptoms of the disease and not its etiology. This may be because it is only in recent decades that it has been proved that the degeneration of the intervertebral disc is a chronic inflammatory disease. Understanding this different concept about the etiology of this pathology is the only way to offer a solution to those who suffer from it. The most recent publications propose treatments geared towards oxidative stress and chronic inflammation. This proves it is not necessary to turn to surgical treatments with fixation of segments on the spine, because this kind of therapeutic approach does nothing to resolve the etiology of the disease. Modulating inflammation with different pharmaceutical resources is an option that is currently being considered.

Medical ozone enables regulating the IVD inflammation at its source, thus modulating oxidative stress, which triggers the inflammatory cascade. Furthermore, it has been demonstrated that there are different epigenetic factors that can change the phenotype of macrophages. We believe that ozone might also be one of these epigenetic factors that intervene in macrophage polarization, enabling them to shift from an inflammatory phase to a repair phase in the same way it would happen with the EDH. As mentioned above ozone reduces MMPs, and increases antioxidant enzymes, stimulates anti-inflammatory cytokines, and secretes anabolic factors. However, we know that although with medical ozone we can control oxidative stress and IVD inflammation, it is not possible to revert the damage. That is why it is necessary to look into new strategies that combine ozone to prepare the microenvironment of the intervertebral space with progenitor cells that enable regeneration of the damaged tissues. 

Progenitor cell implant is a promising method to treat this disease in experimental and clinical models. The IVD being an immune privileged organ, it is possible to use stem cells. Nevertheless, it has been shown that the chances of success of these cells in the implant site are usually very low. This occurs because the degenerated IVD has a hostile microenvironment, with scarce nutrients, low pH, high mechanical pressure, oxidative stress and inflammation. Therefore, the implanted cells have hardly a chance to survive. Numerous methods have been tried in order to improve this situation: genetic modifications, pretreatment of cells, hydrogel materials, etc. 

Reducing oxidative stress and apoptosis could modulate the microenvironment and seems to be a promising, simple and effective therapeutic target [78]. Medical ozone has an antioxidant effect and modulates the immune response, specifically focused on the activated macrophages allowing the implanted cells survival. This local Immune modulation may help the CEP, subchondral bone and vertebral body repair, as well as allowing the integration of all these structures [79]. Chen et al. recently documented that during the co-culture of macrophages and bone mesenchymal stem cells (BMSCs), after the macrophages were polarized from M1 to M2, the expression of pro-inflammatory factors TNF-α, IL-1β and CCR7 in BMSCs notably decreased, while the expression of anti-inflammatory factors IL-4, IL-10 and CD206 were markedly increased, the expression of osteogenesis-related molecules was increased, and the alkaline phosphatase activity was enhanced [80]. M2 macrophages can promote matrix mineralization and mesenchymal stem cells (MSCs) osteogenic differentiation. Promotion of macrophages polarization to M2 is expected to efficiently repair the cavities in degenerative CEP, inhibit the growth of nerves and blood vessels into IVD, and, thus, relieve LBP [81]. It has been described that once the tissue has been damaged, phases M1 and M2 occur simultaneously, but the effects of the proinflammatory mediators override the anti-inflammatory mediators. This shows that the body’s repair work is incapable of reverting the ongoing degenerative processes. That is why, although the disease advances slowly, it is irreversible. This biomolecular explanation allows us to understand the clinical evolution of chronic degenerative disc disease. 

It remains an open question whether any model or regenerative medicine technique can overcome the high loads and harsh IVD microenvironment to enable reparative regeneration of the AF [9]. Consequently, whatever its initial clinical presentation, it is essential to treat them with medical ozone in order to stop, or at least decrease, its persistent degenerative evolution. An efficient and decisive therapeutic intervention is necessary to revert the failed attempt at biological repair. For us ozone represents the most promising epigenetic factor to date.

## 7. Conclusions

Medical ozone is a useful therapeutic resource in degenerative disc disease. According to our vision, until now the fundamentals of the action mechanisms of ozone in this pathology have been insufficiently explained.

Currently, we know that local chronic oxidative stress is one of the most important etiological factors of IVDD. 

It is widely demonstrated in the literature that ozone modulates oxidative stress and inflammation, through the stimulation of Nrf2, and the inhibition of nfkβ a key regulator of inflammation. This is why medical ozone can be used as etiological treatment of IVDD. In this way, it could stop the evolution of the IVDD, which is a degenerative, chronic, and progressive disease. 

Moreover, it has been shown that macrophages can modify its phenotype from M1 (inflammatory phenotype) to M2 (anti-inflammatory phenotype) according to the epigenetic conditions of the environment. Additionally, medical ozone could be an epigenetic factor which could achieve this switch. 

However, medical ozone cannot regenerate damaged tissue, so it is necessary to combine it with the regenerative capacity of progenitor cells obtained from mesenchymal stem cells.

Ozone modulates the hostile microenvironment of the degenerated intervertebral disc, allows implantation and survival of the progenitor cell, and enhances its regenerative capacity.

Medical ozone combined with the progenitor cells could repair and regenerate tissues in the IVDD.

## Figures and Tables

**Figure 1 ijms-24-06538-f001:**
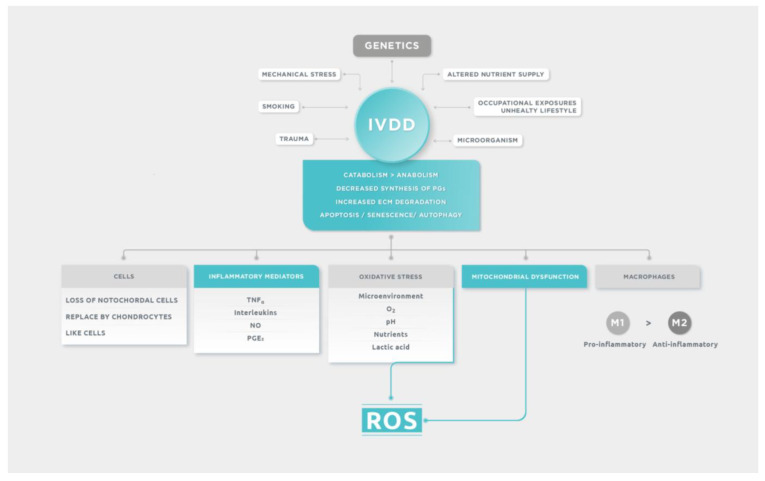
Etiology and main degenerative events in IVDD.

**Table 1 ijms-24-06538-t001:** Characteristics, function and site of production of Proinflammatory and anti-inflammatory cytokines [30].

**Cytokines**	**Characteristic**	**Site of Production**	**IVDD**	**Function**
TNF-α	Pro-inflammatory	Schwann cells Macrophages Mast cells Neutrophils.	Increased	Pain-inducing factor Promotes a cascade of immune reactions and cytokine release
IL-1 β	Pro-inflammatory	Macrophages, Monocytes Dendritic cells	Increased	Increases excitability of neurons. Stimulates ECM degradation
IL-6	Pro-inflammatory	Mast cells Macrophages Lymphocytes Neurons Glial cells	Increased	Decreases pain threshold Increases excitability of neurons. Induces the aggregation of inflammatory cells Activates the release of inflammatory mediators
IFN-ƴ	Pro-inflammatory	Th1 cells Astrocytes Damaged neurons	Increased	Spontaneous pain Pain hypersensitivity. Chronic pain states.
IL-10	Anti-inflammatory	T cells B cells Macrophages, Mast cells	Reduced	Inhibits the release IL-1β, IL-6, and TNF-α
TGF-β	Anti-inflammatory	Activated T and B cells	Increased	Inhibits IL-1β, IL-6, and TNF-α Expression of endogenous opioids. Cell survival, proliferation and differentiation. Inhibits apoptosis.
IL-4	Anti-inflammatory	T cells Mast cells Granulocytes	Increased	Suppress the expression of IL-1β, IL-6, and TNF-α Activation of B cells Differentiates T cells into the Th2 Suppress the activation of macrophages

## Data Availability

No new data were created or analyzed in this study. Data sharing is not applicable to this article.

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
