# Peer review of "The Use of Medical Ozone in Chronic Intervertebral Disc Degeneration Can Be an Etiological and Conservative Treatment"

_ijms, 2023, doi:10.3390/ijms24076538_

Round 1
Reviewer 1 Report
Please reply or modify text according my suggestions. Minor changes only.

Author Response
Dear reviewer
Thank you very much for reviewing our paper.
We have followed your suggestions, and also modified following other reviewers’ comments.
Below we answer some of your concerns.
As you suggested, we have been reviewing bibliography about ozone and its role on MMPs.
Please, consider that the lines that you marked us may be modified by including the corrections of all the reviewers. In this file we place the original lines that you pointed out.
Thank you again for your comments and suggestions.
Best regards.
Line 326: never consolidates
This is a personal observation with my 52 years of experience as a spinal neurosurgeon and thousands of patients with this evolution.
The intervertebral disc disease never consolidates because it goes from an acute stage to a chronic stage, which we call chronic degenerative disc disease.
The extruded portion of the NP can be completely reabsorbed, when it is a sequestration. When the extruded portion is attached to the NP of the intervertebral space, a hernia remnant remains outside the limit of the intervertebral space.
Line 345-350
We add ref 68
Wei B, Zhao Y, Li W, Zhang S, Yan M, Hu Z and Gao B (2022), Innovative immune mechanisms and antioxidative therapies of intervertebral disc degeneration. Front. Bioeng. Biotechnol. 10:1023877. doi: 10.3389/fbioe.2022.1023877
Line 360:
Ok. Done
Line 366:
This is why ozone could promote an improvement
Line 383:
We have consulted the role of ozone on MMPs as the reviewer suggested. More specific in articular cartilage, because it is related to our topic.
The sentence was removed.
It has been described that medical ozone decreased phagocytosis of polymorphonuclear (PMN) in healthy cows but increased in cows with mastitis and milk fever [74]. Moreover, Fernandez-Cuadros et al. [75] describes that ozone also reduces MMPs which have a catabolic effect on articular cartilage demonstrating that medical ozone modulates inflammation.
We add this cite: [75] Fernández-Cuadros, M.E.; Pérez-Moro, O.S.; Albaladejo-Florín, M.J.; Tobar-Izquierdo, M.M.; Magaña-Sánchez, A.; JiménezCuevas, P.; Rodríguez-de-Cía, J. Intra Articular Ozone Modulates Inflammation and Has Anabolic Effect on Knee Osteoarthritis: IL-6 and IGF-1 as Pro-Inflammatory and Anabolic Biomarkers.
Line 428:
The same as above. We add a sentence.
We believe that ozone might also be one of these epigenetic factors that intervene in macrophage polarization, enabling them to shift from an inflammatory phase to a repair phase in the same way it would happen with the EDH. As mentioned above ozone reduces MMPs, and increases antioxidant enzymes, stimulates anti-inflammatory cytokines, and secretes anabolic factors.
Line 468:
Ok. Done
Line 475:
OK. Done

Reviewer 2 Report
1) Review articles are a crowded field and it is standard practice to point out where a new one fits in among its peers with respect to overlap and years covered. Although ozone therapy is included in the title, line 15 makes it clear that what is being reviewed is the biomolecular mechanisms which cause IVDD, with the understanding that ozone meets all therapeutic requirements for these.
The other named reviews discussed in the text are references 17 and 18 with respect to causes in section 3.1 and references 39 and 40 with respect to therapies in 4.1. What is needed in the introduction is thus a summary of what a reader can expect to get out of this review which can't be obtained elsewhere. The named reviews, and other appropriate references such as 44 and 65, should be collected in a table with columns for their unifying themes, main conclusions, and findings from the current review with respect to each theme, such as diagnostics/treatment and restorative/reconstructive strategies (with criticism of ozone section 10.1) for reference 40.
2) All reviews are biased, authors have a vested interest in the field covered. The problem with this review is that the first sentence in the conclusion pronouncing ozone a therapeutic resource was already decided on lines 15-18. What was chosen to review was intended to highlight ozone as a therapy. The resulting conclusion is not surprisingly pretty much what is stated in every ozone therapy study.
Addressing criticism in a review doesn't necessarily result in damage to the research field. Arguments advanced by critics of medical ozone are that studies often involve flawed experimental designs or small sample sizes. Rigourous clinical procedures, such as described reference 59, should be given a separate section.
3) Over half of the references with ozone in the title are more than five years old which gives the risk that the review is becoming outdated already in the manuscript stage. What is needed is a critical evaluation of recent studies with an emphasis on those demonstrating sound clinical practices. Without newer references, the overlap regarding ozone therapy with, for example, the overview reference 60 from 2015, may be substantial.
It can be hoped that more recent studies will have higher quality than what was concluded in C Sconza et al, Oxygen-ozone therapy for the treatment of low back pain: a systematic review of randomized controlled trials, European Review for Medical and Pharmacological Sciences, 25 (2021) 6034-6046, where none of the fifteen evaluated studies conducted 2005-2020 scored a rating of good.
4) Regulatory aspects have no bearing on the specific review subject but they are not without significance to research conducted in the field. FDA's stance that "Ozone is a toxic gas with no known useful medical application in specific, adjunctive, or preventive therapy" is an extreme position, but the more liberal EU goes only as far as legal recognition of medical ozone generators.
Toxicity can't be denied, but the drug Chymopapain (Chymodiactin) has a history that may be of interest for ozone's regulatory fortunes. Approved by the FDA in 1982 for pain relief by dissolution of part of the disc based on success rates similar to ozone therapy, it was discontinued in 2002 for business reasons but attempts have been made to re-instate its production, still with FDA approval. A comparison between the two treatments including the long-terma aspect as described in R J Maciunas and B M Onofrio, The long-term results of chymopapain. Ten-year follow-up of 268 patients after chemonucleolysis, Clin Orthop Relat Res. 206 (1986) 37-41, may be of value.
Although there doesn't seem to be any more recent work, J Buric et al., Five and Ten Year Follow-up on Intradiscal Ozone Injection for Disc Herniation, The International Journal of Spine Surgery, 8 (2014) 17, concluded that benefits were retained after ten years.
Author Response
1) We have carried out an update on the biomolecular mechanisms that cause chronic degenerative disc disease, with particular interest in oxidative stress, inflammation and the role of macrophages.
Although there is an extensive bibliography on the use of medical ozone in spinal diseases, the action mechanisms of ozone in them are under constant study and revision. In this paper, we propose action mechanisms of ozone on oxidative stress and inflammation that are part of this pathology, highlighting that it can be treated etiologically and not only as pain medicine.
We understand your comments and we noticed that our paper structure did not satisfy your expectations.
We wrote in the abstract the synthesis and the novelties that the reader can obtain. This journal allows some flexibility when writing papers like ours in special supplements.
2) The reviewer is right. We have a vested interest in ozone. The article was submitted to the Special Issue "Ozone in Medicine: Recent Advances on the Molecular Mechanisms Underlying Its Biological and Medical Action."
We don´t try to change the results of other research groups. We propose a different explanation of the action mechanisms by which these results are obtained.
We understand your observations. We should have included a section where these issues are addressed, emphasizing that there is rigorous work such as the one in reference 59.
3) In reference 60 it is proposed as an action mechanism of ozone in disc herniation “..ROS, reacting with the proteoglycans of nucleus pulposus, lead to their breakdown, resulting in matrix degeneration with progressive shrinkage and disappearance of the herniated material. Such a reduction of the mechanical irritation lowers the sensitivity of axons”.
In our previous paper (1), we described an action mechanism of ozone different from the dehydration of NP by oxidation of proteoglycans in extruded disc herniation.
The pathophysiology of DDD is different from extruded disc herniation. This is what we detail in this paper and in the previous one (1).
Being a spinal neurosurgeon with 52 years of experience, I see that it is very usual to confuse the pathophysiology of spinal diseases. Disc herniation is frequently misdiagnosed. That is why in this paper we make a detailed description of the DDD, as we did with the extruded disc herniation in 2021. There are common and not common action mechanisms of ozone in both of them.
The most current bibliographical references on the action mechanisms of ozone in spinal pathologies do not coincide with our proposal.
The paper by Sconza et. al concludes that: “The analysis of literature revealed overall poor methodologic quality, with most studies flawed by relevant bias. However, OOT has proven to be a safe treatment with beneficial effects in pain control and functional recovery at short to medium term follow-up”. This matches with our personal experience in 21 years of working with ozone. 52 years of professional experience treating spinal diseases within which 21 years studying and applying ozone therapy, can be as valuable as numerous trials.
4) As you say, the regulatory aspects are not related to the specific topic review. Following your argument and given that this paper is being submitted to the special supplement of medical ozone, we did not include them.
We agree, the FDA's position is extreme and, in our opinion, with commercial biases. Reality shows us that medical ozone is a common practice in the EU and North America.
Although chymopapain has been used in the treatment of intervertebral disc herniation with acceptable results, when compared with medical ozone we must highlight two contradictory aspects.
- With the use of chymopapain, anaphylactic reactions, severe neurological sequelae, bleeding and even death occurred.
- Chymopapain is an enzyme derived from the papaya and as such lyses (denatures) the molecular composition of NP.
Medical ozone has a totally opposite biomolecular effect to chymopapain. We describe these mechanisms in our 2021 paper (1). Therefore, medical ozone administered in immunomodulatory concentrations:
- Has no toxic side effects.
- Modulates the immune system, which reabsorbs the herniated portion of the NP by a physiological mechanism: phagocytosis.
- Preserves vitalized NP within the IV space as observed in post-treatment control MRIs.
- Medical ozone does not lyse the NP located within the IV space.
In our previous work (1), published in 2021, we cited Dr. Buric, and we have reached the same conclusions with good results that are sustained in the long term.
(1) Erario, M.d.l.Á.; Croce, E.; Moviglia Brandolino, M.T.; Moviglia, G.; Grangeat, A.M. Ozone as Modulator of Resorption and Inflammatory Response in Extruded Nucleus Pulposus Herniation. Revising Concepts. Int. J. Mol. Sci. 2021, 22, 9946. https://doi.org/10.3390/ijms22189946

Reviewer 3 Report
Line 14 - change "Nucleus Pulposus" to "nucleus pulposus" (no capitals)
Line 38 - change "Intervertebral" to "intervertebral" (no capital)
Line 116 - change "genetical" to "genetic"
Line 130 - check - should this say the intradiscal pressure in the NP is decreased?
Line 338 - check - should read "...immune abnormality plays an important role..."
Line 368 - mention what regenerative therapies are being referred to e.g. PRP, stem cells etc.
Line 397 - this suggests that ozone concentrations over 10ug/ml can be toxic, but that is not correct, it should be clarified - see Velio Bocci's book "ozone a new medical drug" chapter 2.1 where it is noted that therapeutic ozone concentration ranges from 1ug/ml to 100ug/ml. For chronic pain and IVDD the American Academy of ozone therapy recommends 20ug/ml conc and we use in our office 20 - 25ug/ml conc. with great success for various chronic painful inflammatory conditions including for disc pain.
Line 468 - should read "Ozone is the best epigenetic factor to date." or "Ozone is the best epigenetic factor at the time of this writing."
Figure 1 - use larger font, it is hard to read
General comments: it would be helpful to mention the methods of ozone administration that will help treat IVDD, for example intravenous ozone, ozone injected in the interspinous ligament (Dr. Frank Shallenberger's technique), ozone injected into the disc. Dr. Shallenberger's technique is much simpler and less invasive than direct intradiscal injection and still very effective based on our own experience (this may be unpublished). Even though the ozone is not injected into the disc, it will spread through the tissues and can reach the disc.
If the authors have some experience of their own, it will be interesting to share in this publication.
Author Response
Dear reviewer
Thank you very much for reviewing our paper. We have followed your suggestions, and also modified following other reviewers’ comments. Below we answer some of your concerns.
Please, consider that the lines that you marked us may be modified by including the corrections of all the reviewers. In this file we place the original lines that you pointed out.
Line 14 - change "Nucleus Pulposus" to "nucleus pulposus" (no capitals).
OK. Done
Line 38 - change "Intervertebral" to "intervertebral" (no capital).
OK. Done
Line 116 - change "genetical" to "genetic".
OK. Done
Line 130 - check - should this say the intradiscal pressure in the NP is decreased?
Dehydration of IVD leads to a loss of elasticity of the tissue with loss of its cushioning capacity, therefore the pressure inside the intervertebral space increases. That is why there is a decrease in the height of the disc that is compensated by the increase in the horizontal diameters of the intervertebral disc.
Line 338 - check - should read "...immune abnormality plays an important role..."
OK. Done
Line 368 - mention what regenerative therapies are being referred to e.g. PRP, stem cells, exosomes, etc.
OK. Done
Line 397 - this suggests that ozone concentrations over 10ug/ml can be toxic, but that is not correct, it should be clarified - see Velio Bocci's book "ozone a new medical drug" chapter 2.1 where it is noted that therapeutic ozone concentration ranges from 1ug/ml to 100ug/ml. For chronic pain and IVDD the American Academy of ozone therapy recommends 20ug/ml conc and we use in our office 20 - 25ug/ml conc. with great success for various chronic painful inflammatory conditions including for disc pain.
It's true. It was a typing mistake. It was not what we wanted to express. We use the same concentrations that you propose. The sentence was removed to avoid confusion.
Line 468 - should read "Ozone is the best epigenetic factor to date." or "Ozone is the best epigenetic factor at the time of this writing."
OK. Done
Figure 1 - use larger font, it is hard to read
Original figures and tables were sent to the editor assistant in better quality.
General comments: it would be helpful to mention the methods of ozone administration that will help treat IVDD, for example intravenous ozone, ozone injected in the interspinous ligament (Dr. Frank Shallenberger's technique), ozone injected into the disc. Dr. Shallenberger's technique is much simpler and less invasive than direct intradiscal injection and still very effective based on our own experience (this may be unpublished). Even though the ozone is not injected into the disc, it will spread through the tissues and can reach the disc. If the authors have some experience of their own, it will be interesting to share in this publication.
For the treatment of IVDD we have used intradiscal ozone injections, intramuscular injections and subcutaneous injections for the last 21 years. Sometimes, treatments are complemented with ozonated autohemotransfusion or rectal infusion, according to patient`s need.
We have no experience in Dr. Shallenberger technique and following your suggestions we are going to investigate it.
In our next paper (in course) we are going to share our experience and results.
Thank you again for your comments and suggestions.
Best regards.

Reviewer 4 Report
The document demonstrates a good level of expertise of the authors in this field as well as a good scientific level. Moreover, it is well written and easy to read and the authors took into account the guidelines of the journal.
For this reason, due to its interest and scientific quality, I would like to congratulate the authors.
Author Response
Dear Reviewer
Thank you very much for your kind words.
Writing this paper has been an enormous effort for us, we have been working and studying the bibliography exhaustively during last year.
Your words touched our soul.
Thank you very much again.

Reviewer 5 Report
Authors conduct a literature review on the solution of intervertebral disc degeneration with medical ozone. This article provides useful literature collection and discussion on the prevention of common diseases such as low back pain which is caused by intervertebral disc degeneration. I recommend the paper should be published after minor revisions.
Table 1 should be reedited and appeared in the same page.
Fig. 1 displays etiology and main degenerative events in IVDD. However, most vocabularies in Fig. 1 are too blurry and small to read. Please authors carefully modify it.
Please subscript the numbers in the chemical formulas in the full text.
Line 408: “lower back pain” is not necessary.
Lines 469-476: Conclusion section is weak and insufficient amount, resulted in out of proportion to the main content. Please authors strengthen the Conclusion section.
Author Response
Dear reviewer,
Thank you very much for reviewing our paper. We have followed your suggestions, and also modified following other reviewers’ comments. Below we answer some of your concerns. Please, consider that the lines that you marked us may be modified by including the corrections of all the reviewers. In this file we place the original lines that you pointed out.
In this file we place the original lines that you pointed out to us.
Table 1 should be reedited and appeared in the same page.
Original figures and tables were sent to the editor assistant in better quality.
Fig. 1 displays etiology and main degenerative events in IVDD. However, most vocabularies in Fig. 1 are too blurry and small to read. Please authors carefully modify it.
Original figures and tables were sent to the editor assistant in better quality.
Please subscript the numbers in the chemical formulas in the full text.
OK. Done.
Line 408: “lower back pain” is not necessary.
OK. DONE
Lines 469-476: Conclusion section is weak and insufficient amount, resulted in out of proportion to the main content. Please authors strengthen the Conclusion section.
OK. Done.
Medical ozone is a useful therapeutic resource in degenerative disc disease. According to our vision, until now the fundamentals of the action mechanisms of ozone in this pathology have been insufficiently explained.
Currently, we know that local chronic oxidative stress is one of the most important etiological factors of IVDD.
It is widely demonstrated in the literature that ozone modulates oxidative stress and inflammation, through the stimulation of nrf2, and the inhibition of nfkb a key regulator of inflammation. This is why medical ozone can be used as etiological treatment of IVDD. In this way, it could stop the evolution of the IVDD, which is a degenerative, chronic and progressive disease.
Moreover, it has been shown that macrophages can modify its phenotype from M1 (inflammatory phenotype) to M2 (anti-inflammatory phenotype) according to the epigenetic conditions of the environment. Also, medical ozone could be an epigenetic factor which could achieve this switch.
However, medical ozone cannot regenerate damaged tissue, so it is necessary to combine it with the regenerative capacity of progenitor cells obtained from mesenchymal stem cells.
Ozone modulates the hostile microenvironment of the degenerated intervertebral disc, allows implantation and survival of the progenitor cell, and enhances its regenerative capacity.
Medical ozone combined with the progenitor cells can repair and regenerate tissues in the IVDD.
Thank you again for your comments and suggestions.
Best regards.

Round 2
Reviewer 2 Report
1) Free form does entail a required introduction section which is supposed to include that "The current state of the research field should be reviewed carefully and key publications cited." What previous reviews on the subject have covered would seem to fit that description.
Reversing the order in the title would better match the priorities, Chronic intervertebral disc degeneration and how the use of medical ozone can be an etiological and conservative treatment.
2) Included in the introduction is the request to "Please highlight controversial and diverging hypotheses when necessary".
3) Why not delete references to the older ozone studies and rely on your own personal experience to fulfill the use part of the title.
4) Similar to medical ozone, chymopapain has had its supporters: P Le Goff and P Bourgeois, Should we accept to stop using chymopapain nucleolysis? Joint Bone Spine 69 (2002) 241-243.
My comment to the editor will be to publish as is and disregard this review.